# Inconsistencies between human and macaque lesion data can be resolved with a stimulus-computable model of the ventral visual stream

Tyler Bonnen[1]*, Mark AG Eldridge[2]*

[1]Stanford University, Stanford, United States; [2]Laboratory of Neuropsychology, National Institute of Mental Health, National Institutes of Health, Bethesda, United States

**Abstract** Decades of neuroscientific research has sought to understand medial temporal lobe (MTL) involvement in perception. Apparent inconsistencies in the literature have led to competing interpretations of the available evidence; critically, findings from human participants with naturally occurring MTL damage appear to be inconsistent with data from monkeys with surgical lesions. Here, we leverage a 'stimulus-computable' proxy for the primate ventral visual stream (VVS), which enables us to formally evaluate perceptual demands across stimulus sets, experiments, and species. With this modeling framework, we analyze a series of experiments administered to monkeys with surgical, bilateral damage to perirhinal cortex (PRC), an MTL structure implicated in visual object perception. Across experiments, PRC-lesioned subjects showed no impairment on perceptual tasks; this originally led us(Eldridge et al., 2018) to conclude that PRC is not involved in perception. Here, we find that a 'VVS-like' model predicts both PRC-intact and -lesioned choice behaviors, suggesting that a linear readout of the VVS should be sufficient for performance on these tasks. Evaluating these computational results alongside findings from human experiments, we suggest that results from (Eldridge et al., 2018) alone cannot be used as evidence against PRC involvement in perception. These data indicate that experimental findings from human and non-human primates are consistent. As such, what appeared to be discrepancies between species was in fact due to reliance on informal accounts of perceptual processing.

*For correspondence:
bonnen@stanford.edu (TB);
mark.eldridge@nih.gov (MAGE)

Competing interest: The authors declare that no competing interests exist.

## Editor's evaluation

This article contributes to our section on research advances which offers important follow-up information about previously published articles in *eLife*. This advance offers a valuable integration of work across species that contribute to an ongoing debate about the precise role of medial temporal lobe structures in processes supporting perception as well as memory. The work presented herein uses a model of the ventral visual stream to harmonize predictions across species and leads to compelling evidence for more principled predictions about when and how one might expect contributions to performance. Using this approach has allowed the authors to revise the conclusions of previous work and will likely contribute significantly to future work in this area.

## Introduction

Neuroanatomical structures within the medial temporal lobe (MTL) are known to support memory-related behaviors (*Scoville and Milner, 1957*; *Eichenbaum and Cohen, 2004*; *LaRocque and*

**Figure 1.** Formalizing medial temporal lobe (MTL) involvement in visual object perception. (**a**) Perirhinal cortex (PRC) is an MTL structure situated at the apex of the primate ventral visual stream (VVS), located within rhinal cortex (RHC; see inset). (**b**) To formalize PRC involvement in visual object perception, here we leverage a computational model able to make predictions about VVS-supported performance directly from experimental stimuli. Early model layers best fit electrophysiological recordings from early stages of processing within the VVS (i.e. V4; left, gray); later layers best fit later stages of processing from the VVS (i.e. IT; left, green). We approximate VVS-supported performance by extracting responses from an 'IT-like' model layer (center). Our protocol approximates VVS-supported performance (right; green) while human participants nonetheless outperform model/ VVS performance (*Bonnen et al., 2021*; right, purple). (**c**) Given that humans can outperform a linear readout of the VVS, here we schematize the pattern of lesion results that would be consistent with the PRC involvement in perception (left), results that would indicate that non-PRC brain structures are required to outperform the VVS (center), as well as results which indicate that a visual discrimination task is supported by the VVS (i.e. 'non-diagnostic' because no extra-VVS perceptual processing is required).

*Wagner, 2015*). For decades, experimentalists have also observed MTL-related impairments in tasks designed to test perceptual processing (*Suzuki, 2009*; *Baxter, 2009*). These findings centered on perirhinal cortex (PRC), an MTL structure situated at the apex of high-level sensory cortices (*Figure 1a*). Visual impairments were reported following lesions to PRC in humans and other animals, bolstering a perceptual-mnemonic account of perirhinal function (e.g. *Murray and Bussey, 1999*; *Bussey et al., 2002*; *Lee et al., 2005*; *Lee et al., 2006*; *Barense et al., 2007*; *Inhoff et al., 2019*). However, there were also visual experiments for which no impairments were observed following PRC lesions (e.g. *Buffalo et al., 1998a*; *Buffalo et al., 1998b*; *Stark and Squire, 2000*; *Knutson et al., 2012*). In this was, decades of evidence resulted in a pattern of seemingly inconsistent experimental outcomes, with no formal method for disambiguating between competing interpretations of the available data.

One of the central challenges in this experimental literature has been isolating PRC-dependent behaviors from those supported by PRC-adjacent sensory cortex. In the primate, this requires disentangling PRC-dependent performance from visual behaviors supported by the ventral visual stream (VVS; *DiCarlo and Cox, 2007*; *DiCarlo et al., 2012*). Lacking more objective metrics, experimentalists had relied on informal, descriptive accounts of perceptual demands; terms such as 'complexity' and 'feature ambiguity' were intended to characterize those stimulus properties that are necessary to evaluate PRC involvement in visual object perception. However, this informal approach led to conflicting interpretations of the available evidence, without any means to arbitrate between them. For example, the absence of PRC-related deficits in a given study (e.g. *Stark and Squire, 2000*) has led to the conclusion that PRC is not involved in perception (*Suzuki, 2009*), while others argue that stimuli in stimuli from these studies are not 'complex' enough (i.e. can be represented by canonical visual cortices) and so no perceptual deficits are expected (*Bussey and Saksida, 2002*).

In recent years, deep learning computational methods have become commonplace in the vision sciences. Remarkably, these models are able to predict neural responses throughout the primate VVS directly from experimental stimuli: given an experimental image as input, these models (e.g. convolutional neural networks, CNNs) are able to predict neural responses. These 'stimulus-comptable' methods currently provide the most quantitatively accurate predictions of neural responses throughout the primate VVS (*Yamins et al., 2014*; *Khaligh-Razavi and Kriegeskorte, 2014*; *Rajalingham et al., 2018*; *Bashivan et al., 2019*). For

example, early model layers within a CNN better predict earlier stages of processing within the VVS (e.g. V4; *Figure 1b*: left, gray) while later model layers better predict later stages of processing within the VVS (e.g. IT; *Figure 1b*: left, green). We note that there is not a 1–1 correspondence between these models and the primate VVS as they typically lack known biological properties (*Zhuang et al., 2021*; *Doerig et al., 2022*). Nonetheless, these models can be modified to evaluate domain-specific hypotheses (*Doerig et al., 2022*)—for example by adding recurrence (*Kubilius et al., 2018*; *Kietzmann et al., 2019*) or eccentricity-dependent scaling (*Deza and Konkle, 2020*; *Jonnalagadda et al., 2021*).

Recently, *Bonnen et al., 2021* leveraged these 'VVS-like' models to evaluate the performance of PRC-intact/-lesioned human participants in visual discrimination tasks. While VVS-like models are able to approximate performance supported by a linear readout of high-level visual cortex (*Figure 1b*: right, green), human participants are able to out outperform both VVS-like models and a linear readout of direct electrophysiological recordings from the VVS (*Figure 1b*: right, purple). Critically, VVS-like models approximate PRC-lesioned performance. While these data implicate PRC in visual object processing, there remains experimental data collected from non-human primates which have not been formally evaluated. Like the human literature, non-human primate data have been used to both support and refute PRC involvement in perception. Unlike the naturally occurring lesioned in humans, experiments with non-human primates have unparalleled control over the site and extent of PRC lesions—potentially, providing more incisive tests of competing claims over PRC function. As such, characterizing the discrepancies between human and non-human primate data is a critical step toward developing a more formal understanding of PRC involvement in perception.

In order to resolve this cross-species discrepancy, here we formalize perceptual demands in experiments administered to PRC-intact/-lesioned monkeys (*Macaca mulatta*). We draw from data collected by *Eldridge et al., 2018* which provides striking evidence against PRC involvement in perception: *Eldridge et al., 2018* created multiple stimulus sets, allowing for more a fine-grained evaluation of perceptual behaviors than previous, related work (e.g. *Bussey et al., 2003*). Here, we estimate VVS-supported performance on stimuli from *Eldridge et al., 2018* and compare these predictions to PRC-intact and -lesioned choice behaviors. This modeling approach enables us to situate human and macaque lesion data within a shared metric space (i.e. VVS-model performance); as such, previous observations in the human (e.g. *Figure 1b*: right, green) constrain how data from *Eldridge et al., 2018* can be interpreted; critically, to evaluate PRC involvement in perception, the performance of non-lesioned participants must exceed VVS-modeled performance. Given this, supra-VVS performance may be due to PRC-dependent contributions (schematized in *Figure 1c*: left), or for reasons unrelated to PRC function (schematized in *Figure 1c*: middle). However, if VVS-supported performance approximates PRC-intact behavior, no perceptual processing beyond the VVS should be necessary (schematized in *Figure 1c*: right). We refer to stimuli in this category as 'non-diagnostic'.

## Results

We begin with a task-optimized convolutional neural network, pretrained to perform object classification. We estimate the correspondence between this model and electrophysiological responses from high-level visual cortex using a protocol previously reported in *Bonnen et al., 2021*. We summarize this protocol here, but refer to the previous manuscript for a more detailed account. Using previously collected electrophysiological responses from macaque VVS (*Majaj et al., 2015*), we identify a model layer that best fits high-level visual cortex: Given a set of images, we learn a linear mapping between model responses and a single electrode's responses, then evaluate this mapping using independent data (i.e. left-out images). For each model layer, this analysis yields a median cross-validated fit to noise-corrected neural responses, for both V4 and IT. As is consistently reported (e.g. *Schrimpf et al., 2020*), early model layers (i.e. first half of layers) better predict neural responses in V4 than do later layers (unpaired $t$-test: $t(8) = 2.70, P = 0.015$; *Figure 1b*: left, gray), while later layers better predict neural responses in IT, a higher-level region (unpaired $t$-test: $t(8) = 3.70, P = 0.002$; *Figure 1b*: left, green). Peak V4 fits occur in model layer pool3 (noise-corrected $r = 0.95 \pm 0.30$ STD) while peak IT fits occur in con5_1 (noise-corrected $r = 0.88 \pm 0.16$ STD). For ease, in all subsequent analyses we use model responses from a con5_1-adjacent layer, fc6, which has comparable neural fits but a lower-dimensional representation.

Next we compare model, VVS-supported, and human performance within the same metric space: Instead of fitting model responses directly to electrophysiological recordings in high-level visual cortex, as above, here we evaluate the similarity between the performance supported by the model and high-level visual cortex, as well as human performance on these same stimuli. For this comparison, we leverage electrophysiological responses previously collected from macaque IT cortex (*Majaj et al., 2015*), using a protocol originally detailed in *Bonnen et al., 2021*. We independently estimate model and VVS-supported performance on a stimulus set composed of concurrent visual discrimination trials, using a modified leave-one-out cross-validation strategy. We then determine the model-VVS fit over the performance estimates, as developed in *Bonnen et al., 2021* and outlined in Methods. We can compare model performance with both VVS-supported performance and PRC-intact (human, *n* = 297) performance on these same stimuli, using data from *Bonnen et al., 2021*. On this dataset, a computational proxy for the VVS predicts IT-supported performance ($\beta = 0.81$, $F(1, 30) = 13.33$, $P = 4 \times 10^{-14}$; *Figure 1b*, green), while each are outperformed by ($\beta = 0.24$, $t(31) = 9.50$, $P = 1 \times 10^{-10}$; *Figure 1b*: right, purple). These data suggest that while these models are suitable proxies for VVS-supported performance, human performance is able to exceed a linear readout of the VVS.

With these 'VVS-like' models, we turn to analyses of macaque lesion data. First, we extract model responses to each stimulus in all four experiments administered by *Eldridge et al., 2018*. In these experiments, subjects provided a binary classification for each stimulus: 'cat' or 'dog.' Critically, stimuli were composed not only of cats and dogs, but of 'morphed' images that parametrically vary the percent of category-relevant information present in each trial. For example, '10% morphs' were 90% cat and 10% dog. These morphed stimuli were designed to evaluate PRC involvement in perception by creating maximal 'feature ambiguity,' a perceptual quality reported to elicit PRC dependence in previous work (*Bussey et al., 2002*; *Norman and Eacott, 2004*; *Bussey et al., 2006*; *Murray and Richmond, 2001*). On each trial, subjects were rewarded for responses that correctly identify which category best fits the image presented (e.g. 10% = 'cat', 90% = 'dog', correct response is 'dog'). We evaluate data from two groups of monkeys in this study: an unoperated control group (*n* = 3) and a group with bilateral removal of rhinal cortex, which including peri- and entorhinal cortex. We formulate the modeling problem as a binary forced choice (i.e. 'dog' = 1, 'cat' = 0) and present the model with experimental stimuli. We then extract model responses from a layer that corresponds to 'high-level' visual cortex and learn a linear mapping from model responses to predict the category label. For all analyses, we report the results on held-out data (Methods: Determining model performance).

We first evaluate model performance with the aggregate metrics used by the original authors—not on the performance of individual images, but on the proportion of trials within the same 'bin' that are correct. With the original behavioral data, we average performance across images within each morph level (e.g. 10%, 20%, etc.) across subjects in each lesion group (PRC-intact *Figure 2a*, and -lesioned *Figure 2b*). As reported in *Eldridge et al., 2018*, there is not a significant difference between the choice behaviors of PRC-lesioned and -intact subjects (no significant difference between PRC-intact/-lesion groups: $R^2 = 0.00$, $\beta = 0.01$, $F(1, 86) = 0.07$, $P = 0.941$). For each of these experiments, we extract model responses to all stimuli from a model layer that best corresponds to a high-level visual region, inferior temporal (IT) cortex. Using the model responses from this 'IT-like' model layer to each image, we train a linear, binary classification model on the category label of each image (i.e. 'dog' or 'cat') on 4/5th of the available stimuli. We then evaluate model performance on the remaining 1/5th of those stimuli, repeating this procedure across 50 iterations of randomized train–test splits. A computational proxy for the VVS exhibits the same qualitative pattern of behavior as each subject group (*Figure 2c*, model performance across multiple train–test iterations in black). Moreover, we observe a striking correspondence between model and PRC-intact behavior (*Figure 2b*, purple: $R^2 = 0.98$, $\beta = 0.97$, $t(21) = 33.12$, $P = 6 \times 10^{-19}$) as well as -lesioned subjects (green: $R^2 = 0.99$, $\beta = 0.96$, $t(21) = 57.38$, $P = 1 \times 10^{-23}$). Employing the same metric used to claim no significant difference between PRC-lesion/-intact performance, we find no difference between subject and model behavior ($R^2 = 0.00$, $\beta = -0.01$, $F(1, 86) = -0.11$, $P = 0.915$).

We extend our analysis beyond the aggregate morph- and subject-level analyses used by the original authors, introducing a split-half reliability analysis (Methods: Split-half reliability estimates). This enables us to determine if there is reliable choice behavior, for each subject, at the level of individual images. We restrict our analyses to experiments with sufficient data, as this analysis requires multiple repetitions of each image; we exclude experiments 3 ('Masked Morphs') and 4

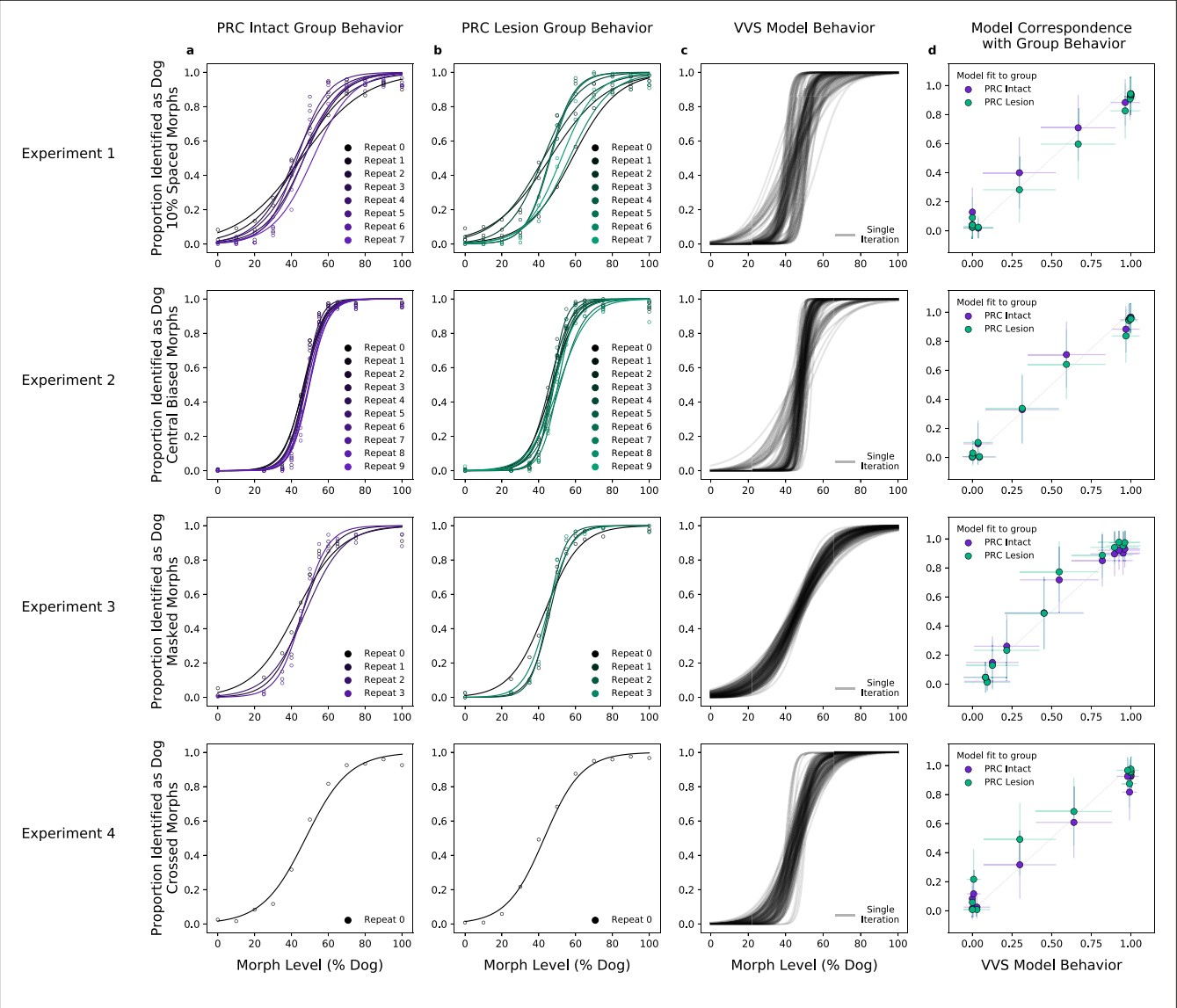

**Figure 2.** A computational proxy for the ventral visual stream (VVS) predicts perirhinal cortex (PRC)-intact and -lesioned behavior. Averaging across subjects and morph levels (i.e. all 10% morphs, 20% morphs, etc.), (**a**) PRC-intact (*n* = 3) and (**b**) PRC-lesioned (*n* = 3) subjects exhibit a similar pattern of responses across experiments (rows 1–4). We present stimuli used in this experiment to a computational proxy for the VVS, extracting model responses from a layer that corresponds with 'high-level' perceptual cortex. From these model responses, we learn to predict the category membership of each stimulus, (**c**) testing this linear mapping on left-out images across multiple train–test iterations (black). (**d**) This computational proxy for the VVS accurately predicts the choice behavior of PRC-intact (purple) and -lesioned (green) grouped subjects (error bars indicate standard deviation from the mean, across model iterations and subject choice behaviors). As such, a linear readout of the VVS appears to be sufficient to perform these tasks, thus there need be no involvement of PRC to achieve neurotypical performance.

The online version of this article includes the following figure supplement(s) for figure 2:

**Figure supplement 1.** Experimental stimuli and protocol from *Eldridge et al., 2018*.

**Figure supplement 2.** Colinearity within the stimulus set revealed by a pixel-level analysis.

**Figure supplement 3.** Pixel-level performance fails on a more conservative evaluation metric.

**Figure supplement 4.** Model approximates primate behavior even with a more conservative evaluation metric.

('Crossed Morphs') due to insufficient repetitions (which can be seen in *Figure 2*, rows 3–4). Across both remaining experiments, we find consistent image-level choice behaviors for subjects with an intact (e.g. median $R^2_{\text{exp1}} = 0.94$, median $R^2_{\text{exp2}} = 0.86$) and lesioned (e.g. median $R^2_{\text{exp1}} = 0.91$, median $R^2_{\text{exp2}} = 0.90$) rhinal cortex (*Figure 3a*: within-subject reliability on the diagonal; PRC-intact subjects

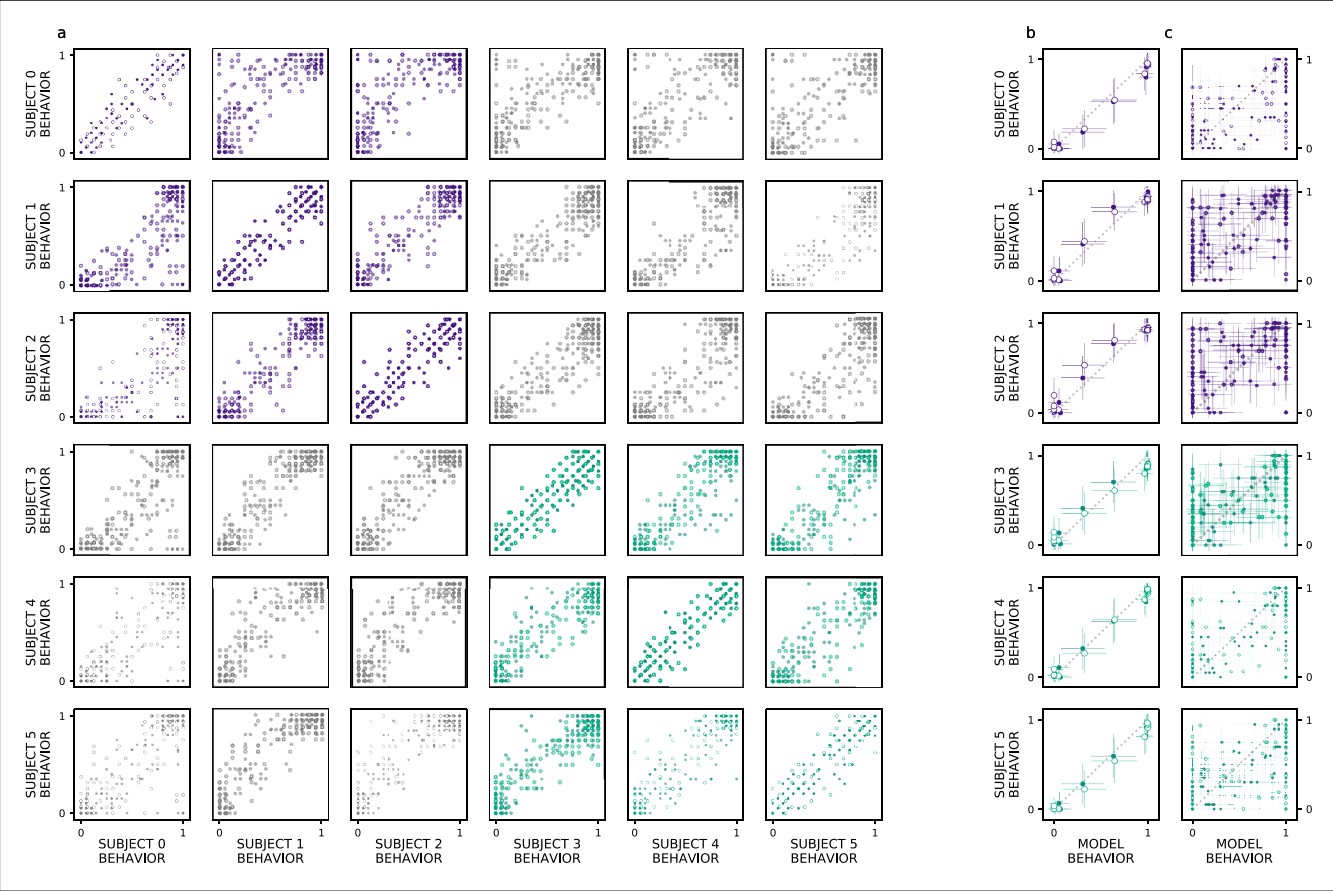

**Figure 3.** Ventral visual stream (VVS) model fits subject behavior for aggregate but not image-level metrics. Here, we perform more granular analyses than those conducted by the authors of the original study: evaluating the model's correspondence with perirhinal cortex (PRC)-lesioned and -intact performance at the level of individual subjects and images. We restrict ourselves to experiments that had sufficient data to determine the split-half reliability of each subject's choice behaviors. First, we determine whether there is reliable image-level choice behavior observed for each subject, that is no longer averaged across morph levels. (**a**) We estimate the correspondence between subject choice behaviors over 100 split-half iterations, for both experiments 1 (closed circles) and 2 (open circles), using $R^2$ as a measure of fit. Each row contains a given subjects' (e.g. subject 0, top row) correspondence with all other subjects' choice behaviors, for PRC-intact (purple) and -lesioned (green) subjects. We find that the image-level choice behaviors are highly reliable both within (on diagonal) and between subjects (off diagonal), including between PRC-lesioned and -intact subjects (gray). We next compare model performance to the behavior of individual subjects, averaging over morph levels in accordance with previous analyses (i.e. averaging performance across all images within each morph level, e.g. 10%). (**b**) We observe a striking correspondence between the model and both PRC-lesioned (green) and PRC-intact (purple) performance for all subjects. (**c**) Finally, for each subject, we estimate the correspondence between model performance and the subject-level choice behaviors, at the resolution of individual images. Although model fits to subject behavior are statistically significant, it clearly does not exhibit 'subject-like' choice behavior at this resolution. Error bars in all experiments indicate standard deviation from the mean, across model iterations and subject choice behaviors.

The online version of this article includes the following figure supplement(s) for figure 3:

**Figure supplement 1.** Ventral visual stream (VVS) model is 'subject-like' for aggregate but not image-level metrics.

in purple, PRC-lesioned subjects in green). We also observe consistent image-level choice behaviors between subjects (e.g. median $R^2_{exp1} = 0.86$, median $R^2_{exp2} = 0.79$). These results indicate there is reliable within- and between-subject variance in the image-by-image choice behaviors of experimental subjects (*Figure 3a*: PRC-intact subjects in purple, PRC-lesioned subjects in green; between-group reliability in gray), suggesting that this behavior is a suitable target to evaluate how well we approximate more granular subject behaviors with a computational proxy for the VVS. We next examine whether the model can predict these more granular, subject- and image-level choice behaviors (see Methods: Consistency estimates).

Our computational approach is able to predict subject-level choice behavior when aggregated across morph levels, for both PRC-intact (e.g. subject 0; $R^2 = 0.99$, $t(21) = 39.30, P = 2 \times 10^{-20}$) and

-lesioned (e.g. subject 4: $R^2 = 0.99$, $t(21) = 45.01, P = 1 \times 10^{-21}$) subjects (*Figure 3b*). Interestingly, the model's fit to subject behavior is indistinguishable from the distribution of between-subject reliability estimates (*Figure 1a*; median of the empirical p(model |reliability$_{\text{between-subject}}$) = 0.592) suggesting that the model exhibits 'subject-like' behaviors at this resolution. Our modeling approach is also able to significantly predict image-level choice behaviors for both PRC-lesioned (e.g. subject 3: $R^2 = 0.86$, $F(1, 438) = 52.79, P = 5 \times 10^{-192}$) and -intact subjects (e.g. subject 1: $R^2 = 0.87$, $F(1, 438) = 53.24, P = 2 \times 10^{-193}$). However, the model behavior is unlikely to be observed under the distribution of between-subject reliability estimates (between-subject reliability distributions visualized in *Figure 1b*; median of the empirical p(model |reliability$_{\text{between-subject}}$) = 0). That is, the model does not exhibit 'subject-like' choice behaviors at the resolution of individual images. This is an important caveat to note when evaluating the correspondence between model performance and animal behavior: as previously reported (*Rajalingham et al., 2018*), even as these models approximate neural responses and choice behaviors in the aggregate (i.e. across images), they do not necessarily capture the trial-by-trial choice behaviors. We elaborate on this further in the discussion.

There are properties of the experimental design in *Eldridge et al., 2018* that encourage a more careful comparison between primate and model behavior. Experimental stimuli contain discrete interpolations between 'cat' and 'dog' images, such that adjacent stimuli within a morph sequence are highly similar (e.g. see *Figure 2—figure supplement 1*). The colinearity in this stimulus set is revealed by running a classification analysis over pixels: a linear readout of stimulus category directly from the vectorized (i.e. flattened) images themselves is sufficient to approximate aggregate performance of all experimental groups ($R^2 = 0.94$, $F(1, 42) = 26.74$, $P = 5 \times 10^{-28}$ ; *Figure 2—figure supplement 2*). To ensure that the VVS modeling approach is not simply a byproduct of the colinearity in the stimuli, we construct a conservative method for model evaluation by restricting training data to images from unrelated morphs sequences (i.e. train on morph sequences A–F, test on morph sequence G). Under this more conservative train–test split, pixels are no longer predictive of primate behavior ($R^2 = 0.05$, $F(1, 42) = 1.42$, $P = 0.164$; *Figure 2—figure supplement 3*), but there remains a clear correspondence between the model and PRC-lesioned ($R^2 = 0.87$, $F(1, 42) = 16.94$, $P = 2 \times 10^{-20}$ ) and -intact performance ($R^2 = 0.88$, $F(1, 42) = 17.39, P = 8 \times 10^{-21}$; *Figure 2—figure supplement 4*). That is, although subjects were able to exploit the colinearity in the stimuli to improve their performance with experience, the correspondence between VVS models and primate choice behaviors is not an artifact of these low-level stimulus attributes.

## Discussion

To evaluate competing claims surrounding PRC involvement in perception, *Eldridge et al., 2018* administered a series of visual classification tasks to PRC-lesioned/-intact monkeys. These stimuli were carefully crafted to exhibit a qualitative, perceptual property that had previously been shown to elicit PRC dependence (i.e. 'feature ambiguity'; *Bussey et al., 2002*; *Norman and Eacott, 2004*; *Bussey*

*et al., 2006*; *Murray and Richmond, 2001*). The absence of PRC-related deficits across four experiments led the original authors to suggest that perceptual processing is not dependent on PRC. Here, we reevaluate this claim by situating these results within a more formal computational framework; leveraging task-optimized convolutional neural networks as a proxy for primate visual processing (*Yamins et al., 2014*; *Rajalingham et al., 2018*; *Schrimpf et al., 2020*). We first determined VVS-model performance on the experimental stimuli in *Eldridge et al., 2018*. We then compared these computational results with monkey choice behaviors, including subjects with bilateral lesions to PRC (*n* = 3), as well as unoperated controls (*n* = 3). For both PRC-lesioned/-intact monkeys, we observe a striking correspondence between VVS model and experimental behavior at the group (*Figure 2d*) and subject level (*Figure 3b*). These results suggest that a linear readout of the VVS should be sufficient to enable the visual classification behaviors in *Eldridge et al., 2018*; no PRC-related impairments are expected.

In isolation, it is ambiguous how these data should be interpreted. For example, if VVS-modeled accuracy was sufficient to explain PRC-intact performance across all known stimulus sets, this would suggest that PRC is not involved in visual object perception. However, previous computational results from humans demonstrate that PRC-intact participants are able to outperform a linear readout of the VVS (schematized in *Figure 1b*: right, purple). Because results from these human experiments are in the same metric space as our current results (i.e. VVS-modeled performance), these data unambiguously constrain our interpretation: for a stimulus set to evaluate PRC involvement in visual processing, participants must be able to outperform a linear readout of the VVS. That is, supra-VVS performance must be observed in order to isolate PRC contributions from those of other possible contributors to these behaviors (e.g. prefrontal cortex, schematized in *Figure 1c*: center). Given that supra-VVS performance is not observed in the current stimulus set (*Figure 2d*; schematized in *Figure 1c*: right), we conclude that experiments in *Eldridge et al., 2018* are not diagnostic of PRC involvement in perception. Consequently, we suggest that these data do not offer absolute evidence against PRC involvement in perception—revising the original conclusions made from this study.

We note that there is meaningful variance in the trial-level behaviors not captured by the current modeling framework. By conducting a more granular analyses than the original study (i.e. an image-level analysis, instead of averaging across multiple images within the same morph level), we found that image-level choice behaviors are reliable both within and between subjects (*Figure 3a*). At this image-level resolution, however, the VVS model does not match the pattern of choice behaviors evident in experimental subjects (*Figure 3c Figure 3—figure supplement 1*). This observation is consistent with previous reports (*Rajalingham et al., 2018*), suggesting that these VVS-like models are best suited to approximate aggregate choice behaviors, not responses to individual images. Many sources of variance have been identified as possible contributors to these subject–model divergences, such as biologically implausible training data (*Zhuang et al., 2021*), or lack of known properties of the primate visual system—for example recurrence (*Kar and DiCarlo, 2020*) or eccentricity-dependent scaling (*Jonnalagadda et al., 2021*).

While admittedly coarse, these computational proxies for the VVS provide an unprecedented opportunity to understand perirhinal function. Their contribution is, principally, to isolate PRC-dependent behaviors from those supported by the VVS. More generally, however, this is possible because these methods directly interface with experimental data—making predictions of VVS-supported performance directly from experimental stimuli, instead of relying on the discretion of experimentalists. This stimulus-computable property of these models provides a formal 'linking function' between theoretical claims with experimental evidence. In turn, this modeling approach creates a unified metric space (in this case, 'model performance') that enables us to evaluate experimental outcomes across labs, across studies, and even across species. We believe that a judicious application of these computational tools, alongside a careful consideration of animal behavior, will enrich the next generation of empirical studies surrounding MTL-dependent perceptual processing.

## Methods
### Evaluating model and VVS-supported performance
We begin with a task-optimized convolutional neural network, pretrained to perform object classification. We estimate the correspondence between this model and electrophysiological responses from

high-level visual cortex using a protocol previously reported in *Bonnen et al., 2021*. We summarize this protocol here, but refer to the previous manuscript for a more detailed account. Using previously collected electrophysiological responses from macaque VVS (*Majaj et al., 2015*), we identify a model layer that best fits high-level visual cortex: Given a set of images, we learn a linear mapping between model responses and a single electrode's responses, then evaluate this mapping using independent data. For each model layer, this analysis yields a median cross-validated fit to noise-corrected neural responses, for both V4 and IT. As is consistently reported (*Rajalingham et al., 2018*; *Yamins et al., 2014*; *Schrimpf et al., 2020*), early model layers (i.e. first half of layers) better predict neural responses in V4 than do later layers (unpaired $t$-test: $t(8) = 2.70, \mathrm{P} = 0.015$; *Figure 1b*: left, gray), while later layers better predict neural responses in IT, a higher-level region (unpaired $t$-test: $t(8) = 3.70, \mathrm{P} = 0.002$; *Figure 1b*: left, green). Peak V4 fits occur in model layer pool3 (noise-corrected $\mathrm{r} = 0.95 \pm 0.30$ STD) while peak IT fits occur in con5_1 (noise-corrected $\mathrm{r} = 0.88 \pm 0.16$ STD).

Next we compare model performance with VVS-supported performance: Instead of fitting model responses directly to electrophysiological recordings in high-level visual cortex, as above, we evaluate the similarity between the performance supported by the model and high-level visual cortex. For this comparison, we again use electrophysiological responses previously collected from macaque IT cortex (*Majaj et al., 2015*), using a protocol detailed in *Bonnen et al., 2021*. We independently estimate model and VVS-supported performance on stimulus set composed of concurrent visual discrimination trials, using a modified leave-one-out cross-validation strategy. We then determine the model-VVS fit over the performance estimates, as developed in *Bonnen et al., 2021*. Each concurrent visual discrimination trial is composed of three images: two images contain the same object$_i$, randomly rotated and projected onto an artificial background; the other image (the 'oddity') contains a second object$_j$, again presented at a random orientation on an artificial background. For each trial, the task is to identify the oddity—that is, the object which does not have a pair—ignoring the viewpoint variation across images.

We use a modified leave-one-out cross-validation strategy to estimate model performance across stimuli in this experiment. For a given sample$_{ij}$ trial, we construct a random combination of three-way oddity tasks to be used as training data; we sample without replacement from the pool of all images of object$_i$ and object$_j$, excluding only those three stimuli that were present in sample$_{ij}$. This yields 'pseudo oddity experiments' where each trial contains two typical objects and one oddity that have the same identity as the objects in sample$_{ij}$ and are randomly configured (different viewpoints, different backgrounds, different orders). These 'pseudo oddity experiments' are used as training data. We reshape all images, present them to the model independently, and extract model responses from an 'IT-like' model layer (in this case, we use fc6 which has a similar fit to IT as conv5_1 but fewer parameters to fit in subsequent steps). From these model responses, we train an L2 regularized linear classifier to identify the oddity across all ($N = 52$) trials in this permutation of pseudo oddity experiments generated for sample$_{ij}$. After learning this weighted, linear readout, we evaluate the classifier on the model responses to sample$_{ij}$. This results in a prediction which is binarized into a single outcome $\{0 \mid 1\}$, either correct or incorrect. We repeat this protocol across 100 random sample$_{ij}$s, and average across them, resulting in a single estimate of model performance for each pair$_{ij}$. To relate model performance with the electrophysiological data, we repeat the leave-one-out cross-validation strategy described above, but in place of the fc6 model representations we run the same protocol on the population-level neural responses from IT and V4 cortex. We perform all analyses comparing model and VVS-supported performance at the object level: for each object$_i$ we average the performance on this object across all oddities (i.e. object$_j$, object$_k$, …) resulting in a single estimate of performance on this item across all oddity tasks ($N = 32$). We can compare model performance with both VVS-supported performance and PRC-intact (human) performance on these same stimuli, using data from *Bonnen et al., 2021*. On this dataset, PRC-intact human behavior outperforms a linear readout of macaque IT (*Figure 1c*: $\beta = 0.24$, $t(31) = 9.50$, $\mathrm{P} = 1 \times 10^{-10}$), while IT significantly outperforms V4 ($\beta = 0.18$, $t(31) = 6.56$, $\mathrm{P} = 2 \times 10^{-7}$). A computational proxy for IT demonstrates the same pattern, predicting IT-supported performance ($\beta = .81$, $\mathrm{F}(1, 30) = 13.33$, $\mathrm{P} = 4 \times 10^{-14}$), outperforming V4 ($\beta = 0.26$, $t(31) = 8.02$, $\mathrm{P} = 5 \times 10^{-9}$), and being outperformed by PRC-intact participants ($\beta = 0.16$, $t(31) = 5.38$, $\mathrm{P} = 7 \times 10^{-6}$).

## Determining model performance

For all estimates of model performance we use a task-optimized convolutional neural network pretrained on Imagenet (*Deng et al., 2009*). For transparency, we report the results from the first instance of this model class used to evaluate these data (*Simonyan and Zisserman, 2014*), but note that these results hold across all model instances evaluated. We preprocess each image from *Eldridge et al., 2018* using a standard computer vision preprocessing pipeline; resizing images to a width and height of 224 × 224, then normalizing each image by the mean ([0.485, 0.456, 0.406]) and standard deviation ([0.229, 0.224, 0.225]) of the distribution of images used to train this model. We present each preprocessed image to the model and extract responses to each image from a layer (fc6) that exhibits a high correspondence with electrophysiological responses to high-level visual cortex (*Bonnen et al., 2021*; and see *Figure 1b*: left). For each experiment, we generate a random train–test split, using 4/5th of the data to train a linear readout (in this case, a logistic regression model). To train this linear readout from model responses, we use an L2-normed logistic regression model implemented in sklearn (*Pedregosa et al., 2011*) to predict the binary category classification (i.e. 'dog' = 1, 'cat' = 0) for each image in the training set. Within the training set, we estimate the optimal regularization strength ('C' from $10^{-5}$ to $10^{-5}$) for the logistic regression model through fivefold cross-validation. We then evaluate model performance on each experiment on independent data (i.e. the remaining 1/5th of stimuli). We repeat this process for 100 permutations (i.e. random 4/5th splits) of stimuli in each condition. Each iteration's model predictions (on independent data) are plotted in *Figure 2c*.

## Consistency estimates

We estimate within- and between-subject consistency using a common protocol. For the given resolution of analysis (either morph- or image level), we require multiple presentations of the same items. For the morph-level analysis, which aggregates stimuli within 'morph levels' (e.g. aggregating across all stimuli that are 0% dog morphs, 10% dog morphs, etc.), all stimulus sets meet this criterion. There are, however, multiple experiments that do not contain sufficient data to perform the image-level analysis, which requires multiple presentations of each stimulus; experiment 4 contains only one presentation of each stimulus, precluding it from our consistency analyses, and experiment 3 contains only four repetitions, which is insufficient for reliable within- and between-subject consistency estimates. Thus, we restrict our consistency estimates to experiments 1 (10 repetitions per image) and 2 (8 repetitions per image).

We estimate all consistency metrics over 100 iterations of random split halves. For each iteration, across all items within a given resolution (where items can refer to either a given morph percent, for the morph-level analysis, or a given image, for the image-level analysis), we randomly split choice behavior into two random splits. In the image-level analysis, for example, for each image $x_i$ within the set of $n$ images, we randomly select half of all trials of $x_i$ (i.e. $x_{i_1}$), and compute the mean of this random sample ($\bar{x}_{i_1}$). We repeat this for all of the $n$ images in this condition (i.e. generating $\bar{x}_{1_1}, \bar{x}_{2_1}, ..., \bar{x}_{n_1}$). We repeat this procedure for the remaining half of trial on each $n$ images (i.e. generating $\bar{x}_{1_2}, \bar{x}_{2_2}, ..., \bar{x}_{n_2}$). Thus, we have two $n$ dimensional vectors, $\vec{v}_1$ and $\vec{v}_2$, where the element in each vector corresponds to a random half of trials drawn from all trials containing that image. We use $R^2$ between these vectors as a measure of fit and repeat this measure over 100 iterations, resulting in a distribution of fits.

For the between-subject consistency metrics, split halves are computed using the same protocol used for the within-subject consistency. For the between-subject analysis, however, $\vec{v}_1$ from subject$_i$s choice behavior is compared to $\vec{v}_2$ from subject$_j$s choice behavior (i.e. we generate a random split from each subject to compare, identical to the within-subject protocol). This approach is an alternative to simply computing the fit between two subjects by aggregating over all collected data. We take this random split approach because when all data are used to compare two subjects, this analysis results in a single-point estimate of the between-subject consistency—not a distribution of values, as is the case in our protocol. This single-point estimate could overestimate the between-subject correspondence, in relation to the within-subject measure. Instead, estimating a random split for each subject and then comparing each subject's data results in a distribution of scores, which provides a measure not only of the average subject–subject correspondence, but also a measure of the variance of the correspondence between subjects (i.e. variation over random splits). Moreover, this approach ensures that both the within- and between-subject correspondence measures are equally powered (i.e. there

are not more samples used to compare between subjects, resulting in a biased estimation of between-subject correspondence).

## Acknowledgements

This work is supported by the Intramural Research Program, National Institute of Mental Health, National Institutes of Health, Department of Health and Human Services (annual report number ZIAMH002032), as well as the National Institute of Neurological Disorders and Stroke of the National Institutes of Health (Award Number F99NS125816), and Stanford's Center for Mind Brain Behavior and Technology. We thank Elizabeth Murray and Anthony Wagner for insightful conversations and suggestions on this manuscript and throughout the course of this work.

## Additional information

### Funding

| Funder | Grant reference number | Author |
|---|---|---|
| National Institute of Mental Health | ZIAMH002032 | Tyler Bonnen |
| National Institute of Neurological Disorders and Stroke | F99NS125816 | Tyler Bonnen |

The funders had no role in study design, data collection, and interpretation, or the decision to submit the work for publication.

### Author contributions

Tyler Bonnen, Conceptualization, Formal analysis, Visualization, Methodology, Writing – original draft, Writing – review and editing; Mark AG Eldridge, Resources, Data curation, Supervision, Methodology, Writing – original draft, Writing – review and editing

### Author ORCIDs

Tyler Bonnen http://orcid.org/0000-0001-8709-1651
Mark AG Eldridge http://orcid.org/0000-0003-4292-6832

### Ethics

All experimental procedures conformed to the Institute of Medicine Guide for the Care and Use of Laboratory Animals and were performed under an Animal Study Protocol approved by the Animal Care and Use Committee of the National Institute of Mental Health, covered by project number: MH002032.

### Decision letter and Author response

Decision letter https://doi.org/10.7554/eLife.84357.sa1
Author response https://doi.org/10.7554/eLife.84357.sa2

## Additional files

### Supplementary files
• MDAR checklist

### Data availability

All scripts used for analysis and visualization can be accessed via github at https://github.com/tzler/eldridge_reanalysis (copy archived at *Bonnen, 2023*). All stimuli and behavioral data used in these analyses can be downloaded via Dryad at https://doi.org/10.5061/dryad.r4xgxd2h7.

The following dataset was generated:

| Author(s) | Year | Dataset title | Dataset URL | Database and Identifier |
|---|---|---|---|---|
| Bonnen T, Eldridge M | 2022 | Data from: Inconsistencies between human and macaque lesion data can be resolved with a stimulus-computable model of the ventral visual stream | https://doi.org/10.5061/dryad.r4xgxd2h7 | Dryad Digital Repository, 10.5061/dryad.r4xgxd2h7 |

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
