## [Editor Report]

This article contributes to our section on research advances which offers important follow-up information about previously published articles in *eLife*. This advance offers a valuable integration of work across species that contribute to an ongoing debate about the precise role of medial temporal lobe structures in processes supporting perception as well as memory. The work presented herein uses a model of the ventral visual stream to harmonize predictions across species and leads to compelling evidence for more principled predictions about when and how one might expect contributions to performance. Using this approach has allowed the authors to revise the conclusions of previous work and will likely contribute significantly to future work in this area.

---

## [Decision Letter]

**Decision letter after peer review:**

[Editors’ note: the authors submitted for reconsideration following the decision after peer review. What follows is the decision letter after the first round of review.]

Thank you for submitting your Research Advance "Inconsistencies between human and macaque lesion data can be resolved with a stimulus-computable model of the ventral visual stream" for consideration by *eLife*. Your article has been reviewed by 2 peer reviewers, and the evaluation has been overseen by a Reviewing Editor and a Senior Editor. The following individual involved in the review of your submission has agreed to reveal their identity: Jonathan Winawer (Reviewer #2).

*Reviewer #1 (Recommendations for the authors):*

This article describes the application of a computational model, previously published in 2021 in Neuron, to an empirical dataset from monkeys, previously published in 2018 in eLife. The 2021 modeling paper argued that the model can be used to determine whether a particular task depends on the perirhinal cortex as opposed to being soluble using ventral visual stream structures alone. The 2018 empirical paper used a series of visual discrimination tasks in monkeys that were designed to contain high levels of 'feature ambiguity' (in which the stimuli that must be discriminated share a large proportion of overlapping features), and yet animals with rhinal cortex lesions were unimpaired, leading the authors to conclude that perirhinal cortex is not involved in the visual perception of objects. The present article revisits and revises that conclusion: when the 2018 tasks are run through the 2021 computational model, the model suggests that they should not depend on perirhinal cortex function after all, because the model of VVS function achieves the same levels of performance as both controls and PRC-lesioned animals from the 2018 paper. This leads the authors of the present study to conclude that the 2018 data are simply "non-diagnostic" in terms of the involvement of the perirhinal cortex in object perception.

The authors have successfully applied the computational tool from 2021 to empirical data, in exactly the way the tool was designed to be used. To the extent that the model can be accepted as a veridical proxy for primate VVS function, its conclusions can be trusted and this study provides a useful piece of information in the interpretation of often contradictory literature. However, I found the contribution to be rather modest. The results of this computational study pertain to only a single empirical study from the literature on perirhinal function (Eldridge et al, 2018). Thus, it cannot be argued that by reinterpreting this study, the current contribution resolves all controversy or even most of the controversy in the foregoing literature. The Bonnen et al. 2021 paper provided a potentially useful computational tool for evaluating the empirical literature, but using that tool to evaluate (and ultimately rule out as non-diagnostic) a single study does not seem to warrant an entire manuscript: I would expect to see a reevaluation of a much larger sample of data in order to make a significant contribution to the literature, above and beyond the paper already published in 2021. In addition, the manuscript in its current form leaves the motivations for some analyses under-specified and the methods occasionally obscure.

– The manuscript does not make a compelling argument as to why Eldridge et al. (2018) is a particularly important example of the prior literature whose reevaluation will change the interpretation of the literature as a whole.

– Considerable effort is expended on evaluating how well the model can "approximate more granular subject behaviors" but it is not explained why this is important, or whether it matters that the model cannot, in fact, approximate image-level subject behavior.

– The section "determining model performance" does not provide sufficient detail for a reader to reproduce the modeling work. The statement that "we estimate the optimal regularization strength for the logistic regression model" appears to be the only statement detailing how the model is trained. This is too sparse and opaque and needs expanding considerably.

– The section "8.2 Consistency estimates" and the caption to Figure S4 both refer to the procedure for estimating the correspondence between subject-subject or subject-model choice behaviors. But these two sections appear to contradict each other. The figure caption says that the authors generate a random split of each subject's data. But in Section 8.2, the last sentence implies (although it's not completely clear) that for the between-subjects metric, all the data from each subject is used. (And it is true that, for a between-subjects analysis, you could use all the data to compute a correlation). Please clarify exactly how the 'split' was generated and whether a split was used for all analyses including between subjects.

*Reviewer #2 (Recommendations for the authors):*

The goal of this paper is to use a model-based approach, developed by one of the authors and colleagues in 2021, to critically re-evaluate the claims made in a prior paper from 2018, written by the other author of this paper (and colleagues), concerning the role of perirhinal cortex in visual perception. The prior paper compared monkeys with and without lesions to the perirhinal cortex and found that their performance was indistinguishable on a difficult perceptual task (categorizing dog-cat morphs as dogs or cats). Because the performance was the same, the conclusion was that the perirhinal cortex is not needed for this task, and probably not needed for perception in general, since this task was chosen specifically to be a task that the perirhinal cortex *might* be important for. Well, the current work argues that in fact the task and stimuli were poorly chosen since the task can be accomplished by a model of the ventral visual cortex. More generally, the authors start with the logic that the perirhinal cortex gets input from the ventral visual processing stream and that if a task can be performed by the ventral visual processing stream alone, then the perirhinal cortex will add no benefit to that task. Hence to determine whether the perirhinal cortex plays a role in perception, one needs a task (and stimulus set) that cannot be done by the ventral visual cortex alone (or cannot be done at the level of monkeys or humans).

There are two important questions the authors then address. First, can their model of the ventral visual cortex perform as well as macaques (with no lesion) on this task? The answer is yes, based on the analysis of this paper. The second question is, are there any tasks that humans or monkeys can perform better than their ventral visual model? If not, then maybe the ventral visual model (and biological ventral visual processing stream) is sufficient for all recognition. The answer here too is yes, there are some tasks humans can perform better than the model. These then would be good tasks to test with a lesion approach to the perirhinal cortex. It is worth noting, though, that none of the analyses showing that humans can outperform the ventral visual model are included in this paper - the papers which showed this are cited but not discussed in detail.

Major strength:

The computational and conceptual frameworks are very valuable. The authors make a compelling case that when patients (or animals) with perirhinal lesions perform equally to those without lesions, the interpretation is ambiguous: it could be that the perirhinal cortex doesn't matter for perception in general, or it could be that it doesn't matter for this stimulus set. They now have a way to distinguish these two possibilities, at least insofar as one trusts their ventral visual model (a standard convolutional neural network). While of course, the model cannot be perfectly accurate, it is nonetheless helpful to have a concrete tool to make a first-pass reasonable guess at how to disambiguate results. Here, the authors offer a potential way forward by trying to identify the kinds of stimuli that will vs won't rely on processing beyond the ventral visual stream. The re-interpretation of the 2018 paper is pretty compelling.

Major weakness:

It is not clear that an off-the-shelf convolution neural network really is a great model of the ventral visual stream. Among other things, it lacks eccentricity-dependent scaling. It also lacks recurrence (as far as I could tell). To the authors' credit, they show detailed analysis on an image-by-image basis showing that in fine detail the model is not a good approximation of monkey choice behavior. This imposes limits on how much trust one should put in model performance as a predictor of whether the ventral visual cortex is sufficient to do a task or not. For example, suppose the authors had found that their model did more poorly than the monkeys (lesioned or not lesioned). According to their own logic, they would have, it seems, been led to the interpretation that some area outside of the ventral visual cortex (but not the perirhinal cortex) contributes to perception, when in fact it could have simply been that their model missed important aspects of ventral visual processing. That didn't happen in this paper, but it is a possible limitation of the method if one wanted to generalize it. There is work suggesting that recurrence in neural networks is essential for capturing the pattern of human behavior on some difficult perceptual judgments (e.g., Kietzmann et al 2019, PNAS). In other words, if the ventral model does not match human (or macaque) performance on some recognition task, it does not imply that an area outside the ventral stream is needed - it could just be that a better ventral model (eg with recurrence, or some other property not included in the model) is needed. This weakness pertains to the generalizability of the approach, not to the specific claims made in this paper, which appear sound.

A second issue is that the title of the paper, "Inconsistencies between human and macaque lesion data can be resolved with a stimulus-computable model of the ventral visual stream" does not seem to be supported by the paper. The paper challenges a conclusion about macaque lesion data. What inconsistency is reconciled, and how?

---

## [Author Response]

[Editors’ note: The authors appealed the original decision. What follows is the authors’ response to the first round of review.]

Reviewer #1 (Recommendations for the authors):This article describes the application of a computational model, previously published in 2021 in Neuron, to an empirical dataset from monkeys, previously published in 2018 in eLife. The 2021 modeling paper argued that the model can be used to determine whether a particular task depends on the perirhinal cortex as opposed to being soluble using ventral visual stream structures alone. The 2018 empirical paper used a series of visual discrimination tasks in monkeys that were designed to contain high levels of 'feature ambiguity' (in which the stimuli that must be discriminated share a large proportion of overlapping features), and yet animals with rhinal cortex lesions were unimpaired, leading the authors to conclude that perirhinal cortex is not involved in the visual perception of objects. The present article revisits and revises that conclusion: when the 2018 tasks are run through the 2021 computational model, the model suggests that they should not depend on perirhinal cortex function after all, because the model of VVS function achieves the same levels of performance as both controls and PRC-lesioned animals from the 2018 paper. This leads the authors of the present study to conclude that the 2018 data are simply "non-diagnostic" in terms of the involvement of the perirhinal cortex in object perception.

We appreciate the Reviewer’s careful reading and synthesis of the background and general findings of this manuscript.

The authors have successfully applied the computational tool from 2021 to empirical data, in exactly the way the tool was designed to be used. To the extent that the model can be accepted as a veridical proxy for primate VVS function, its conclusions can be trusted and this study provides a useful piece of information in the interpretation of often contradictory literature. However, I found the contribution to be rather modest. The results of this computational study pertain to only a single empirical study from the literature on perirhinal function (Eldridge et al, 2018). Thus, it cannot be argued that by reinterpreting this study, the current contribution resolves all controversy or even most of the controversy in the foregoing literature. The Bonnen et al. 2021 paper provided a potentially useful computational tool for evaluating the empirical literature, but using that tool to evaluate (and ultimately rule out as non-diagnostic) a single study does not seem to warrant an entire manuscript: I would expect to see a reevaluation of a much larger sample of data in order to make a significant contribution to the literature, above and beyond the paper already published in 2021. In addition, the manuscript in its current form leaves the motivations for some analyses under-specified and the methods occasionally obscure.

We believe that our comments below outline our rationale for focusing our current analysis on data from Eldridge et al. In brief, these data provide compelling evidence against PRC involvement in perception, and are the only such data with PRC-lesioned/-intact macaques that we were able to secure the stimuli for. As such, data from Eldridge et al. provide a singular opportunity to address discrepancies between human and macaque lesion data. For this reason, we propose the current work as a Research Advance Article type, building off of a manuscript that was previously published in eLife.

– The manuscript does not make a compelling argument as to why Eldridge et al. (2018) is a particularly important example of the prior literature whose reevaluation will change the interpretation of the literature as a whole.

As the Reviewer implies, there are multiple visual discrimination experiments administered to PRC-lesioned and -intact monkeys. We offer two reasons here why we have chosen to focus our analyses on Eldridge et al. 2018.

First, Eldridge et al. 2018 is currently the only relevant visual discrimination experiment administered to PRC-lesioned/-intact macaques for which we have been able to secure experimental stimuli. Prior to the submitting the current manuscript, we solicited authors of the following studies used to support and *refute* PRC involvement in visual perception:

Buffalo, E. A., Ramus, S. J., Clark, R. E., Teng, E., Squire, L. R., & Zola, S. M. (1999). Dissociation between the effects of damage to perirhinal cortex and area TE. Learning & Memory, 6(6), 572-599.Buckley, M. J., Booth, M. C., Rolls, E. T., & Gaffan, D. (2001). Selective perceptual impairments after perirhinal cortex ablation. Journal of Neuroscience, 21(24), 9824-9836.Bussey, T. J., Saksida, L. M., & Murray, E. A. (2002). Perirhinal cortex resolves feature ambiguity in complex visual discriminations. European Journal of Neuroscience, 15(2), 365-374.Bussey, T. J., Saksida, L. M., & Murray, E. A. (2006). Perirhinal cortex and feature-ambiguous discriminations. Learning & Memory, 13(2), 103-105Bussey, T. J., Saksida, L. M., & Murray, E. A. (2003). Impairments in visual discrimination after perirhinal cortex lesions: testing ‘declarative’ vs. ‘perceptual‐mnemonic’ views of perirhinal cortex function. European Journal of Neuroscience, 17(3), 649-660.Eldridge, M. A., Matsumoto, N., Wittig, J. H., Masseau, E. C., Saunders, R. C., & Richmond, B. J. (2018). Perceptual processing in the ventral visual stream requires area TE but not rhinal cortex. elife, 7, e36310.

After reaching out to the original authors, we were only able to secure stimuli from Buckley et al. 2001 and Eldridge et al. 2018. However, the Buckley et al. stimuli have previously been modeled/published by Bonnen, Yamins, and Wagner, 2021 (identical stimuli were later used in human PRC lesioned experiments, with consistent results). As such, we believe that modeling stimuli from Eldridge et al. provide the only novel contribution to the field.

Second, we believe that findings from Eldridge et al. provide the most incisive test of the cross-species discrepancies between human and macaque lesion studies. Currently, the human lesion data have been shown to be entirely consistent PRC involvement in visual object perception (Bonnen, Yamins, and Wagner, 2021). Unlike other macaque stimulus sets we have access to (i.e. Buckley et al. 2001) data from Eldridge et al. were used to refute accounts of PRC involvement in perception. As such, the relative impact of accounting for data in Eldridge et al. is far greater than the majority of studies present in the literature. Moreover, because the design used in Eldridge et al. is different from those used in the human lesion experiments previously modeled (including Buckley et al. 2001) this experiment provides a powerful proof of principle that this modeling framework is useful to understand PRC involvement in perception not only across species, but also across experimental designs.

– Considerable effort is expended on evaluating how well the model can "approximate more granular subject behaviors" but it is not explained why this is important, or whether it matters that the model cannot, in fact, approximate image-level subject behavior.

In order to highlight the logic to these analyses, we have provided a brief explanation in the Results section and clarified our ideas in the discussion. In sum, we hope that this modeling approach will be useful to future experimentalists, and so would like to make clear what the limitations are on predicting animal behaviors. That is, this analysis is not directly relevant to claims about perirhinal function, but more of a methodological claim about model abilities.

From the Results section:

“Our computational approach is able to predict subject-level choice behavior when aggregated across morph levels, for both PRC-intact (e.g. subject 0; R2 = 0.99 β = 1.01, t(21) = 39.30, P = 2 x 10−20) and -lesioned (e.g. subject 4: R2 = 0.99 β = 1.01, t(21) = 45.01, P = 1 x 10−21) subjects (Figure 3b). Interestingly, the model’s fit to subject behavior is indistinguishable from the distribution of between-subject reliability estimates (Figure 1a; median of the empirical P(model|reliability_between-subject_) = 0.592) suggesting that the model exhibits ‘subject-like’ behaviors at this resolution. Our modeling approach is also able to significantly predict image-level choice behaviors for both PRC-lesioned (e.g. subject 3: R2 = 0.86 β = 0.81, F(1, 438) = 52.79, P = 5 x 10−192) and -intact subjects (e.g. subject 1: R2 = 0.87 β = 0.88, F(1, 438) = 53.24, P = 2 x 10−193). However, the model behavior is unlikely to be observed under the distribution of between-subject reliability estimates (between-subject reliability distributions visualized in Figure 1b; median of the empirical P(model|reliability_between-subject_) = 0). That is, the model does not exhibit ‘subject-like’ choice behaviors at the resolution of individual images. This is an important caveat to note when evaluating the correspondence between model performance and animal behavior: as previously reported (Rajalingham et al., 2018), even as these models approximate neural responses and choice behaviors in the aggregate (i.e. across images), they do not necessarily capture the trial-by-trial choice behaviors. We elaborate on this further in the discussion.”

From the Discussion section:

“We note that there is meaningful variance in the trial-level behaviors not captured by the current modeling framework. By conducting a more granular analyses than the original study (i.e. an image-level analysis, instead of averaging across multiple images within the same morph level), we found that image-level choice behaviors are reliable both within- and between-subjects (Figure 3a). At this image-level resolution, however, the VVS model does not match the pattern of choice behaviors evident in experimental subjects (Figure 3c; Supplemental Figure 1b). This observation is consistent with previous reports (Rajalingham et al., 2018), suggesting that these VVS-like models are best suited to approximate aggregate choice behaviors, not responses to individual images. Many sources of variance have been identified as possible contributors to these subject-model divergences, such as biologically implausible training data (Zhuang et al., 2021), or lack of known properties of the primate visual system—e.g. recurrence (Kar and DiCarlo, 2020) or eccentricity-dependent scaling (Jonnalagadda et al., 2021).”

– The section "determining model performance" does not provide sufficient detail for a reader to reproduce the modeling work. The statement that "we estimate the optimal regularization strength for the logistic regression model" appears to be the only statement detailing how the model is trained. This is too sparse and opaque and needs expanding considerably.

We appreciate this request for clarification and agree that this section was not sufficiently clear. We have expanded our description in this section as outlined below:

“For all estimates of model performance we use a task-optimized convolutional neural network pretrained on Imagenet (Deng et al., 2009). For transparency, we report the results from the first instance of this model class used to evaluate these data (Simonyan and Zisserman, 2014), but note that these results hold across all model instances evaluated. We preprocess each image from Eldridge et al., 2018 using a standard computer vision preprocessing pipeline; resizing images to a width and height of 224x224, then normalizing each image by the mean ([0.485, 0.456, 0.406]) and standard deviation ([0.229, 0.224, 0.225]) of the distribution of images used to train this model. We present each preprocessed image to the model and extract responses to each image from a layer (fc6) that exhibits a high correspondence with electrophysiological responses to high-level visual cortex (Bonnen et al., 2021; and see Figure 1b: left). For each experiment, we generate a random train-test split, using 4/5th of the data to train a linear readout (in this case, a logistic regression model). To train this linear readout from model responses, we use a l2-normed logistic regression model implemented in sklearn (Pedregosa et al., 2011) to predict the binary category classification (i.e. ‘dog’=1, ‘cat’=0) for each image in the training set. Within the training set, we estimate the optimal regularization strength (‘C’ from 10−5 to 10−5) for the logistic regression model through 5-fold cross validation. We then evaluate model performance on each experiment on independent data (i.e. the remaining 1/5th of stimuli). We repeat this process for 100 permutations (i.e. random 4/5th splits) of stimuli in each condition. Each iteration’s model predictions (on independent data) are plotted in Figure 2c.”

– The section "8.2 Consistency estimates" and the caption to Figure S4 both refer to the procedure for estimating the correspondence between subject-subject or subject-model choice behaviors. But these two sections appear to contradict each other. The figure caption says that the authors generate a random split of each subject's data. But in Section 8.2, the last sentence implies (although it's not completely clear) that for the between-subjects metric, all the data from each subject is used. (And it is true that, for a between-subjects analysis, you could use all the data to compute a correlation). Please clarify exactly how the 'split' was generated and whether a split was used for all analyses including between subjects.

We appreciate this request for clarification and agree that this section was not sufficiently clear. We have added a few sentences to the “consistency estimates” section in order to clarify that the splits for the within- and between-subject analyses are generated in an identical manner, as well as including some of the rationale behind this decision:

*“*We estimate within- and between-subject consistency using a common protocol. For the given resolution of analysis (either morph- or image-level), we require multiple presentations of the same items. For the morph-level analysis, which aggregates stimuli within ‘morph levels’ (e.g. all stimuli that are designed to be 0% dog, 10%, etc.), all stimulus sets meet this criterion. There are, however, multiple experiments that do not contain sufficient data to perform the image-level analysis, which requires multiple presentations of each stimulus; experiment four contains only one presentation of each stimulus, precluding it from our consistency analyses, and experiment three contains only 4 repetitions, which is insufficient for reliable within- and between-subject consistency estimates. Thus, we restrict our consistency estimates to experiments one (10 repetitions per image) and two (8 repetitions per image).

We estimate all consistency metrics over 100 iterations of random split-halves. For each iteration, across all items within a given resolution (where items can refer to either a given morph percent, for the morph-level analysis, or a given image, for the image-level analysis), we randomly split choice behavior into two random splits. In the image-level analysis, for example, for each image x_i_ within the set of n images, we randomly select half of all trials of x_i_ (i.e.xi1), and compute the mean of this random sample (xi1). We repeate this for all of the n images in this condition (i.e. generating x¯11,x¯21...,x¯n1). We repeat this procedure for the remaining half of trial on each n images (i.e. generating x¯12,x¯22,...,x¯n2). Thus, we have two n dimensional vectors, v→1 and v→2, where the element in each vector corresponds to a random half of trials drawn from all trial containing that image. We use R^2^ between these vectors as a measure of fit and repeat this measure over 100 iterations, resulting in a distribution of fits. For the between-subject consistency metrics, split halves are computed using the same protocol used for the within-subject consistency. For the between-subject analysis, however, v→1 from subject*_i_*s choice behavior is compared to v→2 from subject*_j_*s choice behavior (i.e. we generate a random split from each subject to compare, identical to the within-subject protocol). This approach is an alternative to simply computing the fit between two subjects by aggregating over all collected data. We take this random split approach because when all data are used to compare two subjects, this analysis results in a single point estimate of the between-subject consistency—not a distribution of values, as is the case in our protocol. This single-point estimate could overestimate the between-subject correspondence, in relation to the within-subject measure. Instead, estimating a random split for each subject and then comparing each subjects’ data results in a distribution of scores, which provides a measure not only of the average subject-subject correspondence, but also a measure of the variance of the correspondence between subjects (i.e. variation over random splits). Moreover, this approach ensures that both the within- and between-subject correspondence measures are equally powered (i.e. there are not more samples used to compare between subjects, resulting in a biased estimation of between-subject correspondence).”

Reviewer #2 (Recommendations for the authors):The goal of this paper is to use a model-based approach, developed by one of the authors and colleagues in 2021, to critically re-evaluate the claims made in a prior paper from 2018, written by the other author of this paper (and colleagues), concerning the role of perirhinal cortex in visual perception. The prior paper compared monkeys with and without lesions to the perirhinal cortex and found that their performance was indistinguishable on a difficult perceptual task (categorizing dog-cat morphs as dogs or cats). Because the performance was the same, the conclusion was that the perirhinal cortex is not needed for this task, and probably not needed for perception in general, since this task was chosen specifically to be a task that the perirhinal cortex *might* be important for. Well, the current work argues that in fact the task and stimuli were poorly chosen since the task can be accomplished by a model of the ventral visual cortex. More generally, the authors start with the logic that the perirhinal cortex gets input from the ventral visual processing stream and that if a task can be performed by the ventral visual processing stream alone, then the perirhinal cortex will add no benefit to that task. Hence to determine whether the perirhinal cortex plays a role in perception, one needs a task (and stimulus set) that cannot be done by the ventral visual cortex alone (or cannot be done at the level of monkeys or humans).There are two important questions the authors then address. First, can their model of the ventral visual cortex perform as well as macaques (with no lesion) on this task? The answer is yes, based on the analysis of this paper. The second question is, are there any tasks that humans or monkeys can perform better than their ventral visual model? If not, then maybe the ventral visual model (and biological ventral visual processing stream) is sufficient for all recognition. The answer here too is yes, there are some tasks humans can perform better than the model. These then would be good tasks to test with a lesion approach to the perirhinal cortex. It is worth noting, though, that none of the analyses showing that humans can outperform the ventral visual model are included in this paper - the papers which showed this are cited but not discussed in detail.Major strength:The computational and conceptual frameworks are very valuable. The authors make a compelling case that when patients (or animals) with perirhinal lesions perform equally to those without lesions, the interpretation is ambiguous: it could be that the perirhinal cortex doesn't matter for perception in general, or it could be that it doesn't matter for this stimulus set. They now have a way to distinguish these two possibilities, at least insofar as one trusts their ventral visual model (a standard convolutional neural network). While of course, the model cannot be perfectly accurate, it is nonetheless helpful to have a concrete tool to make a first-pass reasonable guess at how to disambiguate results. Here, the authors offer a potential way forward by trying to identify the kinds of stimuli that will vs won't rely on processing beyond the ventral visual stream. The re-interpretation of the 2018 paper is pretty compelling.

We thank the Reviewer for the careful reading of our manuscript and for providing a fantistics synthesis of the current work.

Major weakness:It is not clear that an off-the-shelf convolution neural network really is a great model of the ventral visual stream. Among other things, it lacks eccentricity-dependent scaling. It also lacks recurrence (as far as I could tell).

We agree with the Reviewer completely on this point: there is little reason to expect that off-the-shelf convolutional neural networks should predict neural responses from the ventral visual stream, for the reasons outlined above (no eccentricity-dependent scaling, no recurrence) as well as others (weight sharing is biologically implausible, as well as the data distributions and objective functions use to optimize these models). Perhaps surprisingly, these models *do* provide quantitatively accurate accounts of information processing throughout the VVS; while this is well established within the literature, we were careless to simply assert this as a given without providing an account of these data. We appreciate the Reviewer for making this clear and we have changed the manuscript in several critical ways in order to avoid making unsubstantiated claims in the current version. We hope that these changes also make it easier for the casual reader to appreciate the logic in our analyses.

First, in the introduction, we outline some of the prior experimental work that demonstrates how deep learning models are effective proxies for neural responses throughout the VVS. We also demonstrate this model-neural fit in the current paper using electrophysiological recordings (more on that below), but also including comments about the limitation of these models raised by the Reviewer:

*“*In recent years, deep learning computational methods have become commonplace in the vision sciences. Remarkably, these models are able to predict neural responses throughout the primate VVS directly from experimental stimuli: given an experimental image as input, these models (e.g. convolutional neural networks, CNNs) are able to predict neural responses. These ‘stimulus-comptable’ methods currently provide the most quantitatively accurate predictions of neural responses throughout the primate VVS (Bashivan et al., 2019; Khaligh-Razavi and Kriegeskorte, 2014; Rajalingham et al., 2018; Yamins et al., 2014). For example, early model layers within a CNN better predict earlier stages of processing within the VVS (e.g. V4; Fig. 1b: left, grey) while later model layers better predict later stages of processing 50 within the VVS (e.g. IT; Fig. 1b: left, green). We note that there is not a 1-1 correspondence between these models and the primate VVS as they typically lack known biological properties (Doerig et al., 2022; Zhuang et al., 2021). Nonetheless, these models can be modified to evaluate domain-specific hypotheses (Doerig et al., 2022)—e.g. by adding recurrence (Kietzmann et al., 2019; Kubilius et al., 2018) or eccentricity-dependent scaling (Deza and Konkle, 2020; Jonnalagadda et al., 2021).”

In the introduction we also more clearly demarcate prior contributions from our recent computational work, and highlight how models approximate the performance supported by a linear readout of the VVS, but fail to reach human-level performance:

*“*Recently, Bonnen et al., 2021 leveraged these ‘VVS-like’ models to evaluate the performance of PRC- intact/-lesioned human participants in visual discrimination tasks. While VVS-like models are able to approximate performance supported by a linear readout of high-level visual cortex (Fig. 1b: right, green), human participants are able to out outperform both VVS-like models and a linear readout of direct electrophysiological recordings from the VVS (Fig. 1b: right, purple). Critically, VVS-like models approximate PRC-lesioned performance. While these data implicate PRC in visual object processing,…”

Results from these analyses were essential to understanding the logic of the paper but previously (as noted by the Reviewer) this critical evidence was cited but not directly presented. We include a description to these we describe these data in the introduction more thoroughly, and substantially change figure one, in order to visualize these data (b)

Moreover, we include a over of the methods and data used to generate these plots in the results and methods sections (only showing the results (lines 85-112), for brevity):

While there is little reason to expect that off-the-shelf convolutional neural networks should predict neural responses from the ventral visual stream, we believe that these modifications to the manuscript (to the introduction and figure one, as well as the results and methods sections) make clear that these models are nonetheless useful methods for predicting VVS responses and the behaviors that depend on the VVS.

To the authors' credit, they show detailed analysis on an image-by-image basis showing that in fine detail the model is not a good approximation of monkey choice behavior. This imposes limits on how much trust one should put in model performance as a predictor of whether the ventral visual cortex is sufficient to do a task or not. For example, suppose the authors had found that their model did more poorly than the monkeys (lesioned or not lesioned). According to their own logic, they would have, it seems, been led to the interpretation that some area outside of the ventral visual cortex (but not the perirhinal cortex) contributes to perception, when in fact it could have simply been that their model missed important aspects of ventral visual processing. That didn't happen in this paper, but it is a possible limitation of the method if one wanted to generalize it. There is work suggesting that recurrence in neural networks is essential for capturing the pattern of human behavior on some difficult perceptual judgments (e.g., Kietzmann et al 2019, PNAS). In other words, if the ventral model does not match human (or macaque) performance on some recognition task, it does not imply that an area outside the ventral stream is needed - it could just be that a better ventral model (eg with recurrence, or some other property not included in the model) is needed. This weakness pertains to the generalizability of the approach, not to the specific claims made in this paper, which appear sound.

We could not agree more with the Reviewer on these points. It *could have* been the case that these models' lack of correspondence with known biological properties (e.g. recurrence) led them to lack something important about VVS-supported performance, and that this would derail the entire modeling effort here. Surprisingly, this has not been the case, as is evident in the clear correspondence between model performance and monkey data in Eldridge et al. 2018. Nonetheless, we would expect that other experimental paradigms should be able to reveal these model failings. And future work evaluating PRC involvement in perception must contend with this very problem in order to move forward with this modeling framework. That is, it is of critical importance that these VVS models and the VVS itself exhibit similar failure modes, otherwise it is not possible to use these models to isolate behaviors that may depend on PRC.

A second issue is that the title of the paper, "Inconsistencies between human and macaque lesion data can be resolved with a stimulus-computable model of the ventral visual stream" does not seem to be supported by the paper. The paper challenges a conclusion about macaque lesion data. What inconsistency is reconciled, and how?

It appears that this point was lost in the original manuscript; we have tried to clarify this idea in both the abstract and the introduction. In summary, the cumulative evidence from the human lesion data suggest that PRC is involved in visual object perception, while there are still studies in the monkey literature that suggest otherwise (e.g. Eldridge et al. 2018). In this manuscript, we suggest that this apparent inconsistency is, in fact, simply a consequence of reliance on information interpretations of the monkey lesion data.

We have made substantive changes to the abstract so this is an obvious, central claim.

We have also made substantive changes to the introduction to make resolving this cross-species discrepancy a more central aim of the current manuscript, (lines 56-83)